# Photodynamic Therapy Using 5-Aminolevulinic Acid (Ala) for the Treatment of Chronic Periodontitis: A Prospective Case Series

**Dorina Lauritano** [1,*,†], **Giulia Moreo** [2,*,†], **Annalisa Palmieri** [3], **Fedora Della Vella** [4], **Massimo Petruzzi** [4], **Daniele Botticelli** [5] **and Francesco Carinci** [1]

1   Department of Translational Medicine, University of Ferrara, 44121 Ferrara, Italy; francesco.carinci@unife.it
2   Department of Biomedical, Surgical and Dental Sciences University of Milan, 20122 Milan, Italy
3   Department of Experimental, Diagnostic and Specialty Medicine, University of Bologna, 40126 Bologna, Italy; annalisa.palmieri@unibo.it
4   Interdisciplinary Department of Medicine, University of Bari, 70121 Bari, Italy; dellavellaf@gmail.com (F.D.V.); massimo.petruzzi@uniba.it (M.P.)
5   Private practice, 47921 Rimini, Italy; info@ariminum.eu
*   Correspondence: dorina.lauritano@unife.it (D.L.); moreo.giulia@gmail.com (G.M.)
†   These authors contributed equally to this work.

**Featured Application: This paper highlights the advantages of photodynamic therapy for treatment of periodontal disease.**

**Abstract:** Aim: The objective of this study was to compare the efficacy of supportive periodontal therapy (i.e., scaling and root planning, SRP) alone versus ALADENT medical device used in association with SRP in the treatment of chronic periodontitis in adult patients. Materials and Methods: A total of 20 patients with a diagnosis of chronic periodontitis (40 localized chronic periodontitis sites) aged between 35 and 55 were selected. None of these patients previously received any surgical or non-surgical periodontal therapy, and they presented radiographic evidence of moderate bone loss. Two non-adjacent sites in different quadrants were identified and observed in each patient, analyzing treatment effectiveness (split-mouth design). Clinical pocket depth, clinical attachment loss, and bleeding on probing were evaluated at time 0 and after 6 months, while microbial analysis (MA) was conducted at baseline and after 15 days. Significant differences were calculated using SPSS program and paired simple statistic *t*-test. Results: Total bacteria loadings had a statistically significant reduction before and after treatment with SRP (left site) (total average decrease of 27%). The sites treated with SRP plus ALADENT (right) showed a significantly reduced total bacterial loading compared to the untreated sites (right) (total average decrease of 75%). Mean values of CAL/PD and percentages data of BOP, recorded after SRP + ALADENT therapy, showed a higher reduction (CAL = 2.42, PD = 2.87 mm, 90% of sites with no bleeding) than those obtained after SRP treatment (CAL = 4.08 mm, PD = 4.73 mm, 70% of sites with no bleeding). Conclusion: The treatment of moderate and severe chronic periodontitis should include, beside SRP, the use of ALADENT medical device, which has been proved to be a useful adjuvant therapy.

**Keywords:** periodontal disease; chronic periodontitis; moderate and severe periodontitis; adjuvant periodontal treatment; photodynamic therapy; 5-aminolevulinic acid; bactericidal activity; deactivation and detoxification of periodontal pathogens; scaling and root planning

## 1. Introduction

Periodontal disease (PD) is highly prevalent worldwide. Approximately 50% of the adults present PD [1–3]. PD can be defined as chronic multi-factorial inflammatory diseases, depending on the composition of dental plaque biofilm, and characterized by the deterioration of tooth-supporting tissues [4,5]. The American Academy of Periodontology

(AAP) and the European Federation of Periodontology (EFP) provided a new classification scheme for periodontal and peri-implant diseases during the 2017 World Workshop, identifying three forms: Necrotizing periodontal disease, periodontitis as a manifestation of systemic condition, and periodontitis. Regarding this last group, four stages (based on severity and complexity of management) were identified: Initial (stage I) and moderate (stage II) periodontitis, severe periodontitis with potential for additional tooth loss (stage III) and with potential for loss of the dentition (stage IV). The risk of disease progression was classified into slow (grade A), moderate (grade B), and rapid (grade C) [6–12]. Gingivitis and periodontitis represent two different conditions, showing different phenotypes: Gingivitis, the mildest form of PD, presents itself as a reversible gingival inflammation, it arises due to the presence of bacterial biofilm deposits, and it is characterized by the absence of periodontal attachment apparatus destruction [13,14]. The combination of bacterial infection and persistent inflammatory response can progressively destroy the deeper periodontal tissues, leading to periodontitis, which is represented by the loss of alveolar bone and clinical attachment and by the onset of periodontal pocket [15]. The progression of periodontitis causes tooth mobility and, if not treated, can result in tooth exfoliation. Periodontal inflammation may induce low-grade systemic inflammation, damaging the function of other organs [16,17]. Periodontitis represents a condition that occurs because of the chronic inflammatory process proper to gingivitis, although it is essential to note that gingivitis does not necessarily progress to periodontitis [18,19]. The onset of periodontitis is associated with the presence of periodontal pathogenic microorganisms, individual genetic factors, immune pathways and lifestyle, which may enable bacteria to express their pathogenic potential [20,21].

Periodontal treatment aims to avoid the progression of periodontitis by resolving oral infection and thus preserving the natural dentition, restoring tissues that have been damaged, and preventing its recurrence [22–24]. These goals may be achieved by eliminating supragingival and subgingival biofilms: As literature widely reported, the prevention of periodontal disease may be achieved by performing non-surgical therapy (scaling and root planning) and maintaining adequate oral hygiene [25–29]. However, SRP alone does not permit to control the disease since it could only provide a temporary change of the subgingival bacteria, maintaining the presence of periodontopathic microorganisms and, therefore, allowing the recolonization of the periodontal sites. Limitations of SRP may derive from the presence of bone defects, limited access to the periodontal tissues, and complex tooth anatomy [30,31]. Adjunctive use of new therapies can provide a further advantage in managing the disease: The literature showed that laser therapy can improve the healing of treated sites [32–35].

## 2. Photodynamic Therapy

Photodynamic therapy (PDT) can be considered as a valid alternative in the managing of PD, thus representing an important novelty in the treatment of oral infections. PDT can occur with different mechanisms: It oxidates the cellular components, such as plasma membranes and DNA, and photo-activates the endogenous or exogenous photosensitizers (PS), forming reactive species and leading to cellular death [36–38]. The photodynamic process consists of a photosensitizer (PS) reaction caused by a light source with a specific wavelength that can be absorbed and that, therefore, activates the photosensitizer itself [39,40]. The PS, when irradiated, passes from its lowest energy level to the short-lived excited singlet state, which can be transformed into the long-lived excited triplet state. The triplet PS, in the presence of ambient oxygen, can produce highly reactive singlet oxygen and toxic reactive oxygen species (ROS) [41]. ROS are able to penetrate into cell membranes, inducing oxidation of membrane lipids and amino acids, agglutination of proteins, and oxidative damage of nucleic acids [42,43]. In recent years, research has identified some photosensitizing substances. Some of them have an artificial origin (methylene blue, toluidine blue) [44]. 5-aminolevulinic acid (ALA) is the sensitizing substance that characterizes this medical device. ALA is a homologous compound already present in our body as an integral part of

the metabolism process of the EME group [45,46]. For over ten years, ALA photodynamic therapy was experimented with, and it produced new specific vehicles mainly addressed to oncological dermatology, demonstrating significant efficacy, selectivity, and a very high safety profile. Applications have confirmed the extreme effectiveness of ALA medical devices towards the broadest spectrum of pathogens [47,48]. These experiences preceded and evolved in the formula of the Aladent medical device (ALADENT, Alphastrumenti Srl., Melzo (MI), Italy) for the treatment of periodontitis and peri-implantitis.

Our research aimed to analyze the effectiveness of a new medical device (ALADENT), a patented (PCT/IB2018/060368, 19 December 2018) thermosetting gel with 5% ALA content, used as an adjuvant therapy together with SRP in the treatment of chronic periodontitis in adult patients. Several articles have already highlighted the in-vitro efficacy of ALADENT against pathogens involved in periodontal disease [49–51].

## 3. Materials and Methods

A total of 20 patients with a diagnosis of moderate and severe chronic periodontitis (stage II and III of the 2017 World Workshop classification of periodontal disease, on the basis of management complexity and severity) were randomly selected. Inclusion criteria were as follows: (1) Diagnosis of moderate and severe periodontitis according to the 2017 classification by AAP and EFP [12], (2) presence of two non-adjacent sites belonging to different quadrants for which the periodontal treatment was indicated, (3) age range from 35 to 55, (4) periodontal sites did not receive any surgical or non-surgical periodontal therapy prior to the enrollment in the study. The patients were excluded from the study if they met any of the following criteria: (1) Pregnancy, (2) a history of taking antibiotics or using antibacterial mouth rinses for the past 6 months, (3) a history of systemic disease, (4) history of psychiatric disorders, (5) teeth with furcation involvement, (6) teeth with previous traumatic lesions, (7) smoking, and drug or alcohol abuse. During the sample selection, all 67 patients, aged between 35 and 55, who came to our clinic for a first visit or periodic monitoring between January and December 2019 were taken into consideration. Initially, 31 patients, diagnosed with moderate and severe chronic periodontitis, were recruited: 6 of them were not selected for our study since they recently received surgical periodontal treatment (2 subjects) or non-surgical therapy (4 patients). 2 subjects were smokers, 1 woman was pregnant, 2 patients had a history of systemic disease and they were consequently excluded from the research. Subjects enrolled in the study (20 patients) were given a detailed verbal description of the procedure and signed consent forms.

## 4. Clinical Methods

A total of 20 patients (i.e., 40 sites) were selected and grouped into two categories: Control and test (split mouth design). The control group (20 left sites) was treated with SRP. The test group was treated by SRP plus ALADENT (20 right sites).

One clinician (D.L.) performed the diagnosis of chronic periodontitis of the enrolled sample. The diagnosis was assessed following Multi-State Markov Models proposed by Mdala et al. in 2014 [52], which associate the state of chronic periodontitis to clinical pocket depth (PD)/clinical attachment loss (CAL) values higher than 4 mm with or without BOP. For this reason, prior to treatment, PD, bleeding on probing (BOP) and CAL were detected in 6 different points for each selected site: Disto-lingual/palatal, mesio-lingual/palatal mesio-buccal, disto-buccal, mid-lingual/palatal, and mid-buccal. BOP was calculated using the Papillary Bleeding Index (PBI) introduced by Saxer and Muhlemann in 1975 [48], identifying four different scores, depending on the gravity of gingival papillae bleeding. Microbial analysis was also conducted: The isolation of the sites was obtained by using cotton rolls, subgingival samples were collected with sterile absorbable paper points (size 60) in the deepest part of the periodontal pocket and were immediately processed in a microbiological lab. Total bacterial loadings (TBL) were then evaluated.

The same clinician (D.L.) performed SRP treatment in all the selected sites at the baseline measurement, using ultra sonic scalers. After SRP, PDT with ALADENT was

performed in the test sites by the same operator. The product was inserted in each pocket with the dedicated capillary tip. The incubation of the product lasted 60 min. Each site was illuminated with a 630 nm led source (TL-01 Alpha Strumenti) for 7 min, with a total light dose of 120 J/cmq (Figure 1). The tip of the illuminator was kept very close to the site, and it was positioned perpendicular to the surface of the gingival mucosa. TBL and PD/CAL/BOP were recorded again from both sites in each patient after 15 days and after 6 months, respectively. After treatment, all the included patients were asked to observe adequate oral hygiene maneuvers (both in case and control sites): Teeth brushing three times a day, daily use of dental floss, application of periodontal gel with 1% chlorhexidine. Principal outcomes were reported using means and percentages.

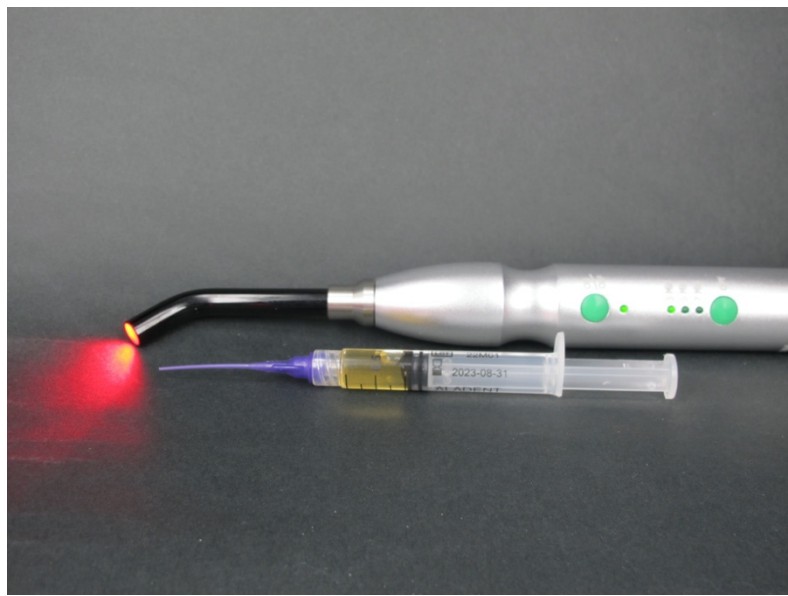

**Figure 1.** Aladent thermosetting gel and TL-01 630 nm led activator.

Total bacterial loading was analyzed by performing Real-Time Polymerase Chain Reaction [53].

## 5. Statistical Analysis

Statistically significant results were obtained by using SPSS program and paired simple statistic *t*-test.

## 6. Results

### 6.1. Microbiological Results

Compared to T0 (time before treatment), total bacteria loadings have a statistically significant reduction after 15 days of treatment with SRP (left sites) (Table 1 and Figure 2) (total average decrease of 27%). In the sites treated with SRP plus ALAD (right), the total bacterial loading was significantly reduced compared to the untreated sites (left) (total average decrease of 75%) (Table 2).

**Table 1.** In right (R-SRP plus ALADENT) and left (L-SRP) sites.

|  |  | Media | N | Standard Deviation | Standard Error Media |
|---|---|---|---|---|---|
| Couple 1 | TBL-0-R | 159,278.2 | 20 | 113,470.1862 | 25,372.705 |
|  | TBL-1-R | 39,819.7 | 20 | 14,683.3434 | 3283.2954 |
| Couple 2 | TBL-0-L | 184,440.1 | 20 | 98,658.8991 | 22,060.8005 |
|  | TBL-1-L | 134,641.2 | 20 | 111,791.5357 | 24,997.3473 |

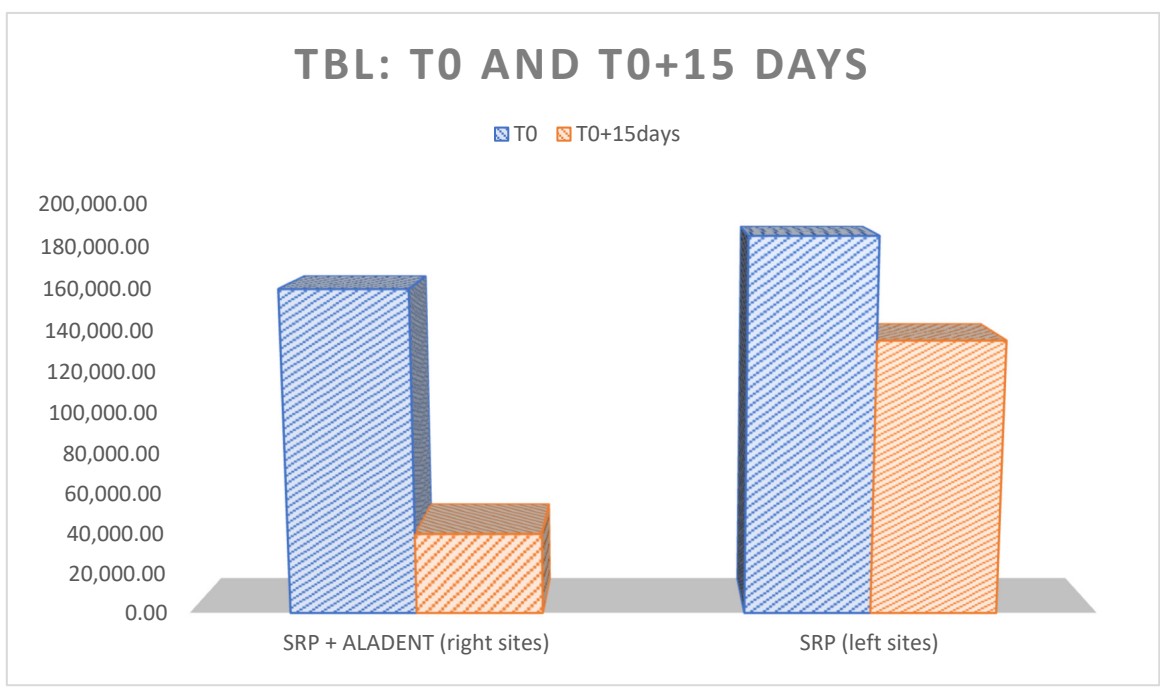

**Figure 2.** Total bacteria loading (TBL) before (time 0) and two weeks after treatment (time 0 + 15 days) in right (**R**) (SRP) and left (**L**) (SRP plus ALADENT) sites.

**Table 2.** Total bacterial loading (TBL) reduction in the sites treated with SRP (left-L) compared to the site treated with SRP plus ALADENT (right-R).

|  |  | Media | Standard Deviation | Standard Error Media | Sig (2-Code) |
|---|---|---|---|---|---|
| Couple 1 | TBL-0-R—TBL-1-R | 119,458.5 | 107,599.2581 | 24,059.9255 | 0.001 |
| Couple 2 | TBL-0-L—TBL-1-L | 49,798.8 | 25,907.0034 | 5792.9821 | 0.001 |

*6.2. Clinical Results*

Mean values of CAL and PD, recorded after SRP + ALADENT therapy, highlighted a higher reduction than those obtained after SRP treatment (Table 3, Figures 3–5). Before treatment, CAL and PD mean values of case sites were equal to 4.65 and 5.14 mm respectively, while after treatment, a mean value of 2.42 and 2.87 mm was recorded for the two indicators. Control sites presented, at time 0, a CAL of 4.76 mm and a PD equal to 5.24 mm, after 6 months, PD and CAL mean values were 4.73 mm and 4.08 mm respectively. With regards to BOP (Figure 6), the majority of the cases and controls sites (70%, 28 sites out of 40) presented, before treatment, several isolated bleeding points or a single line of blood (Score 2), while Score 1 (single discreet bleeding point) and Score 3 (interdental triangle fills with blood shortly after probing) were found in 10% and 20% of the selected sites, respectively. After SRP plus ALADENT treatment, 90% of the sites showed no bleeding (Score 0), compared to the SRP treated sites, that mostly reached Score 1.

**Table 3.** CAL and PD values after treatment with SRP alone and SRP plus ALADENT: Results are statistically significant at * *p* value < 0.01.

|  | CAL (Mean ± SD) | PD (Mean ± SD) | *p* Value |
|---|---|---|---|
| SRP (20 sites) | 4.08 ± 0.44 | 4.73 ± 0.34 | <0.00001 * |
| SRP plus Aladent (20 sites) | 2.42 ± 0.46 | 2.87 ± 0.43 | <0.00001 * |

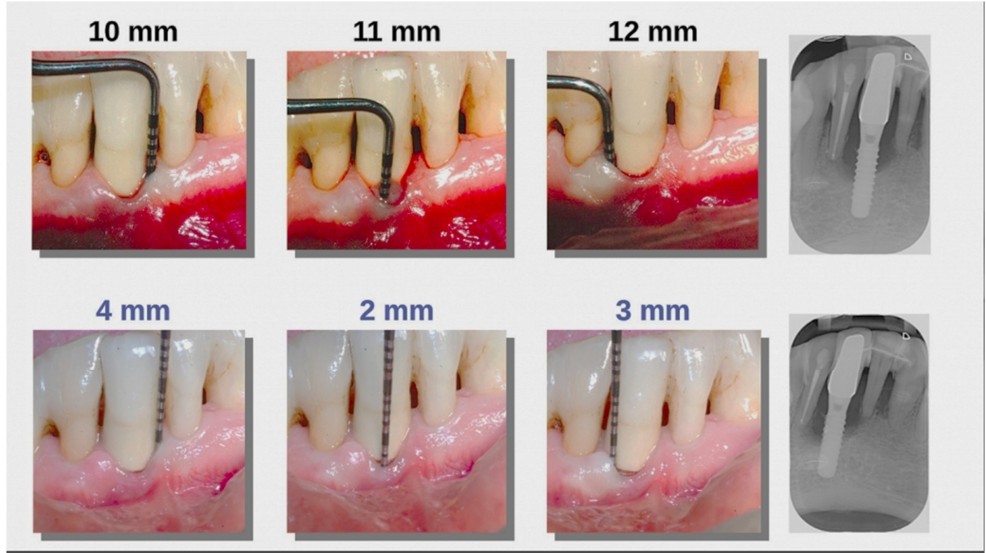

**Figure 3.** Examples of pocket depth reduction before and after treatment with Aladent.

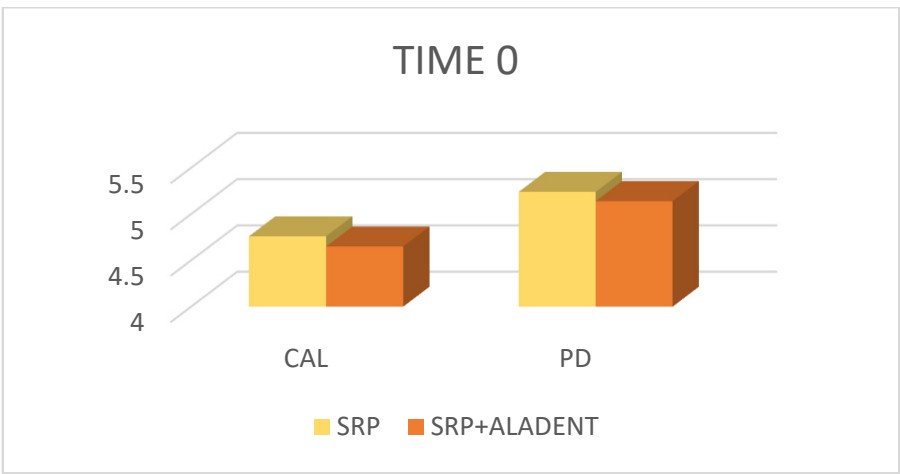

**Figure 4.** CAL and PD mean values (mm) of the selected sites before treatment with SRP alone and SRP plus ALADENT.

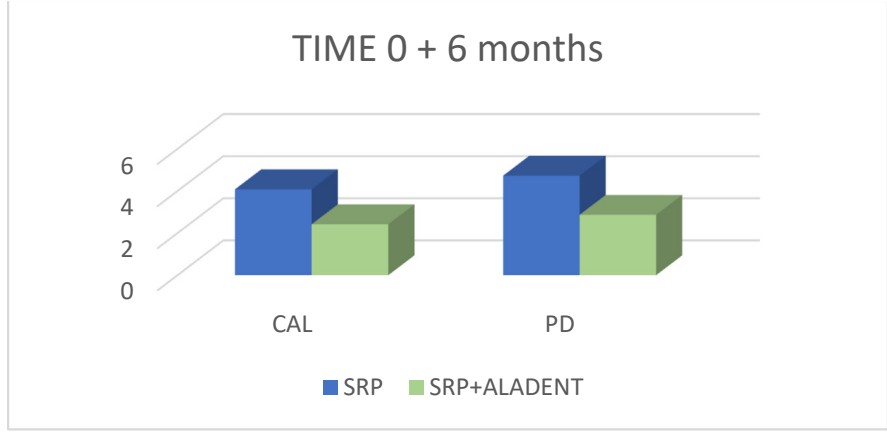

**Figure 5.** CAL and PD mean values (mm) of the selected sites after treatment with SRP alone and SRP plus ALADENT.

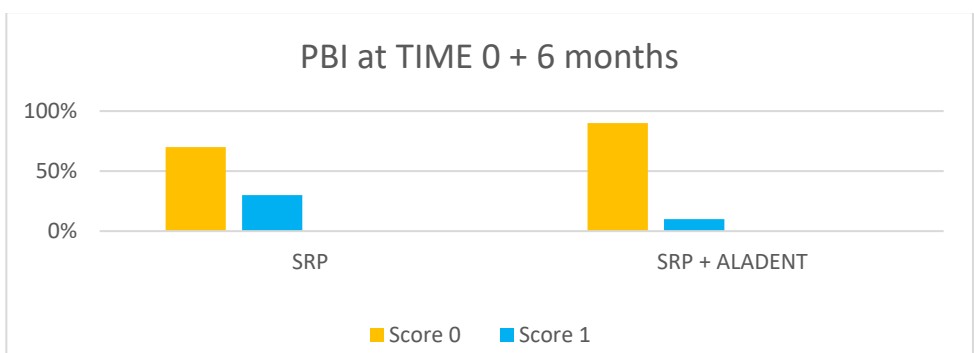

**Figure 6.** PBI of the selected sites after treatment with SRP alone and SRP plus ALADENT.

## 7. Discussion

Periodontal therapy is successful when it provides the decrease of the periodontal bacteria in the subgingival area [54]. *Porphyromonas gingivalis*, *Treponema denticola*, and *Tannerella forsythia*, included in the red complex bacteria, represent the main periodontal pathogens, which can induce periodontal disease progression: Significant periodontal tissue destruction, increasing depth of periodontal pockets and bleeding [31,55]. The literature reported that subgingival debridement (scaling and root planning) leads to a temporary change in the composition of subgingival microorganisms, including species like the microflora found in healthy periodontal sites [56,57]. However, recolonization of the periodontal sites by spirochetes and motile rods one to two months after the end of the therapy, together with the persistence of red and orange complexes bacteria in the deepest part of the periodontal pockets, make it clear that non-surgical periodontal treatment (SRP) is not sufficient in contrasting chronic periodontitis [58–60].

Data obtained in our study demonstrated that the non-invasive procedure of PDT, in association with non-surgical therapy, is effective in the treatment of periodontitis, obtaining the deactivation and detoxification of periodontal pathogens, the decrease of probing depth and clinical attachment loss [61,62]. Microorganism photoinactivation can be obtained by direct or indirect mechanisms. The direct method involves high-energy photons directly damaging vital parts of bacterial cells, while the indirect antimicrobial PDT can be described as an oxygen-dependent photochemical reaction generated by the activation of a PS through a light source. The combination of a non-toxic PS, visible light with appropriate wavelength and oxygen leads to the production of highly reactive and cytotoxic ROS (singlet oxygen), consequently destroying cells [63–68]. The randomized controlled split-mouth clinical trial by Birang et al. [69] evaluated the role of adjunctive PDT in patients suffering from chronic periodontitis, highlighting significant short-term improvements (three months follow-up) in terms of bacterial count, CAL, and PD. Positive long-term clinical effects of PDT were demonstrated in the randomized clinical trial by Alwaeli et al. [70], in which improvement in periodontal clinical indicators (CAL, PD, BOP) of 136 sites was recorded for at least one year. Thirty-three patients were selected in the research by Andersen et al. [71] and divided in three groups, treated with SRP alone, PDT alone, and SRP associated with PDT: The data proved that periodontal sites, which received the combined therapy (scaling and root planning + PDT) showed a higher reduction of clinical attachment loss, probing pocket depth decrease, and BOP decrease in comparison with those treated with SRP alone.

Our in-vivo analysis evaluated the effectiveness of photodynamic therapy, performed by the concurrent use of a novel product containing aminolaevulinic acid (ALA) and red led irradiation against different types of bacteria involved in oral infections. Support periodontal therapy alone is widely used, but data obtained in our research reported a greater decrease of bacterial load when SRP is associated with the administration of ALADENT, which is the sole medical device using a PS applied topically. ALADENT is easy to use and causes no discomfort for the patient. It is administered directly into the periodontal

pocket, drastically reducing the adverse effects that could occur during a systemic therapy. ALADENT presents a bactericidal activity, reducing gingival inflammation and thus representing an effective procedure in the treatment of periodontitis. The aminolaevulinic acid is the precursor of protoporphyrin IX (PpIX), it is normally synthesized in mitochondria, thanks to the condensation reaction between glycine and succinyl-CoA [72], but it can also be administrated exogenously as a prodrug [51]. The ALA molecules are better absorbed intracellularly than the PpIX, and their accumulation and photoactivation in porphyrins (PAPs) is higher in cells with greater metabolic activity, such as tumor cells, bacteria, and inflammatory cells [73]. ALADENT device allows, therefore, to maximize the penetration of the PS in the bacterial membrane. The effectiveness of ALA PDT depends on several factors: Bacteria cell wall composition, pH value, concentration, and duration of exposure to the PS [74–76]. Our study highlighted that ALADENT cannot only be effective in destroying pathogenic microorganisms colonizing the subgingival area but also in significantly improving CAL, PD, and BOP after six months of follow-up. The suppression or eradication of these microbes and the positive effect on periodontal clinical parameters result in periodontal health improvement. Moreover, the novel ALADENT contains the preservatives, potassium sorbate and sodium benzoate, enhancing the bactericidal effect in periodontal pockets [77,78], and it was demonstrated that the use of a red led, irradiating ALA, allows the antimicrobic effect to be longer maintained [79]. Thanks to its properties, the use of ALADENT enhances the control of the periodontal pockets during the non-surgical treatment of periodontitis. It was also demonstrated that ALADENT may significantly reduce the progression of periodontitis and the destruction of periodontal tissues in animal models.

Besides ALADENT, the literature also presented other interesting innovative strategies in the treatment of periodontal disease: The clinical trial by Butera et al. [80] proposed the use of a probiotic-based toothpaste and chewing gum in addition to SRP treatment, demonstrating that it could be a valid support option against periodontal microorganisms, improving several clinical periodontal indexes. Another study by the same author [81] compared the effectiveness of a chlorhexidine-based and postbiotics-based periodontal gel used after SRP, recording a significant reduction of probing pocket depth, recession, dental mobility, and bleeding on probing after both treatments. Probiotics were largely recommended as complementary periodontal therapy since symbiotic bacteria are able to decrease the level of proinflammatory cytokines, potentiating the effect of SRP [82]. Postbiotics derive from microbial fermentation and metabolic activity of microorganisms, providing antiseptic, antioxidant, and anti-inflammatory effects [83]. Finally, beside SRP, the adjunctive treatment with low-level laser therapy (LLLT) obtaining photobiomodulation (PBM) was demonstrated to be more effective in reducing probing pocket depth, bleeding on probing, and plaque index compared to the ozone therapy [84] since it promotes gingival fibroblast proliferation and FGF-b and type-1 collagen expression, improving healing processes through bio-stimulation.

## 8. Conclusions

Our study aimed at finding a more suitable protocol for clinical practice by reducing the treatment duration and costs of materials, and that still had an effective antimicrobial action on periodontal bacteria. This preliminary study showed that ALADENT had a great antibacterial potential on different Gram-positive and Gram-negative bacteria involved in antibiotic resistance and oral infections. Together with bacterial load reduction, ALADENT provides the reduction of periodontal soft tissue inflammatory parameters, such as PD, CAL, and BOP. Therefore, ALADENT can be considered as adjuvant therapy in the management of chronic periodontitis. Besides ALADENT, also other innovative adjuvant periodontal approaches should be further investigated (probiotics, postbiotics, and laser therapies), in order to provide a reliable treatment option against periodontitis in addition to traditional strategies.

**Author Contributions:** Conceptualization, Writing—original draft preparation D.L.; Validation, G.M.; Investigation, D.B.; Data curation, F.D.V. and A.P.; Methodology, M.P.; Supervision, F.C. All authors have read and agreed to the published version of the manuscript.

**Funding:** This research received no external funding.

**Institutional Review Board Statement:** Not applicable.

**Informed Consent Statement:** Informed consent was obtained from all subjects involved in the study.

**Data Availability Statement:** Not applicable.

**Conflicts of Interest:** The authors declare no conflict of interest.

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
