# Peer review of "Photodynamic Therapy Using 5-Aminolevulinic Acid (Ala) for the Treatment of Chronic Periodontitis: A Prospective Case Series"

_applsci, doi:10.3390/app12063102_

Round 1

Reviewer 1 Report

The authors have positively addressed the raised questions. Thus, I recommend the acceptance of this manuscript.

Author Response

Ferrara, 12th March 2022

            Dear Reviewer,

                        I thank you for recommending the acceptance of the paper.

Yours sincerely,

Dorina Lauritano

Reviewer 2 Report

The authors have improved the manuscript according to the comments from reviewers. I think it can be accepted now.

Author Response

(The authors gave the same response as above.)

Reviewer 3 Report

Thank you very much for allowing me to review this manuscript, which is of considerable interest to the dental profession

Add keywords, these are few and not specific

Introduction, to add the new classification of periodontal disease

Materials and methods
Classify patients according to the new classification
Add exclusion criteria
How was the sample number calculated?
What tools were used for professional and home treatment?

Results.
Change the resolution of the images, they are too post produced

Discussion

To be expanded the innovative proactive approaches in addition to photodynamic therapy in periodontal treatment for the clinical and microbiological management for the reduction of the bacterial load of the red and orange complex by laser and ozone, probiotics, postbiotics and postbiotics of the research group of Scribante et al.

Conclusion, to be reformulated with a proactive approach

Bibliography to be expanded

Author Response

Ferrara, 12th March 2022

            Dear Reviewer,

            Many thanks for the insightful comments and suggestions of the referees. We have

made corresponding revisions according to their advice. Words in red are the changes

we have made in the text.

  • Add keywords, these are few and not specific. We added more specific keywords.

  • Introduction, to add the new classification of periodontal disease. We added the new classification of periodontal disease

Materials and methods

  • Classify patients according to the new classification. We classified the included patients according to the new periodontal classification
  • Add exclusion criteria. We added exclusion criteria
  • How was the sample number calculated? We performed a sample size calculation, obtaining a series of cases. We took into consideration the patients in the first visit or patients who came for the periodic monitoring, including the subjects suffering from moderate and severe periodontitis, and selected those who met the inclusion criteria.
  • What tools were used for professional and home treatment? We added the description of home oral hygiene protocols for both case and control sites. ALADENT device and ultrasonic scalers (to perform scaling and root planning) were used as professional tools.

Results.

  • Change the resolution of the images, they are too post-produced. We changed the resolution of the images.

Discussion

  • To be expanded the innovative proactive approaches in addition to photodynamic therapy in periodontal treatment for the clinical and microbiological management for the reduction of the bacterial load of the red and orange complex by laser and ozone, probiotics, postbiotics, and postbiotics of the research group of Scribante et al. We expanded the “Discussion” section, adding research about adjuvant periodontal therapies studied by the research group of Scribante et al.

  • Conclusion, to be reformulated with a proactive approach. We reformulated the “Conclusion” section with a proactive approach.

  • Bibliography to be expanded. We expanded the bibliography.

Thank you for receiving our manuscript and considering it for publication.

We appreciate your time and look forward to your response.

Yours sincerely,

Dorina Lauritano

Round 2

Reviewer 3 Report

The manuscript has been corrected revised, can be published 

This manuscript is a resubmission of an earlier submission. The following is a list of the peer review reports and author responses from that submission.

Round 1

Reviewer 1 Report

-Please provide the ethical committee approval for the study

-Authors explained in the title the prospective nature of the treatment:

Did you perform a sample size calculation?

Who did perform the periodontal examination?

What is the standard deviation of graph 3,4,and 5?

Who performed the treatment?

Who applied the photodynamic therapy? Was it a double-blind treatment or not?

How did you decide what side to be treated with SRP and what with SRP+ALA?

I think there is a great bias in this work: the TBL baseline data (T0) show a significantly higher TBL in test group respect control...so the two groups were not comparable to each other. In order to sustain such conclusions, a proper randomized clinical trial should be performed, with a  proper sample size calculation.

Why did you not insert leged under the tables?

Please substitute commas with points in the decimal numbers

Author Response

Ferrara, 19th February 2022

Dear editor:

Many thanks for the insightful comments and suggestions of the referees. We have

made corresponding revisions according to their advice. Words in red are the changes

we have made in the text.

  • Please provide the ethical committee approval for the study. Our study doesn’t provide for an ethical committee, since it was conducted in a private dental office and since the ALADENT device was already on the market when we started our research.

  • Did you perform a sample size calculation? Yes, we performed a sample size calculation, obtaining a series of cases. We added in the “Materials and Methods” section the description of sample size calculation.

  • Who did perform the periodontal examination? Who performed the treatment? Who applied the photodynamic therapy? Was it a double-blind treatment or not? The periodontal examination and the treatment (SRP and SRP + ALADENT) were performed by the same operator. We specified it in the text (“Clinical Methods” section). It was not a double-blind treatment, our study reported a series of cases.

  • What is the standard deviation of graphs 3,4 and 5? SD of graphs 3 and 4 was reported in Table III. SD of PBI was not calculated, since the outcome measure was a score.

  • How did you decide what side to be treated with SRP and what with SRP+ALA? Both selected sites of each patient were diagnosed with chronic periodontitis, therefore the sites did not present clinical differences. We consequently decided to perform the same treatment in the sites belonging to the same side of the dental arch.

  • I think there is a great bias in this work: the TBL baseline data (T0) show a significantly higher TBL in test group respect control...so the two groups were not comparable to each other. In order to sustain such conclusions, a proper randomized clinical trial should be performed, with a proper sample size calculation. We tested 10 left sites, treating them with SRP+ ALADENT, and 10 sites, treating them with SRP alone. The differences in TBL between the sites are due to the fact that the patients physiologically brush their teeth more to one side than the other. Consequently, the TBL at T0 shows different values.

  • Why did you not insert leged under the tables? We added legends under the tables.

  • Please substitute commas with points in the decimal numbers. We substituted commas with points in the decimal numbers.

Thank you for receiving our manuscript and considering it for publication.

We appreciate your time and look forward to your response.

Yours sincerely,

Prof.ssa Dorina Lauritano

Reviewer 2 Report

This manuscript described a photodynamic therapy for chronic periodontitis in adult patients. The therapy combats the different types of bacteria involved in oral infections by using a novel product containing aminolaevulinic acid (ALA) and red LED irradiation at the same time. After treatment with SRP plus ALADENT, the CAL/PD mean and BOP data showed a statistically significant reduction in total bacterial load, with even 90% of sites no bleeding. The ALADENT not only has bactericidal activity but also is easy to use, injecting it directly into the periodontal pocket can greatly reduce the adverse reactions that may occur during systemic treatment. This manuscript presents a very effective method for the treatment of periodontal disease. Give the high novelty of this work, I recommend the acceptance of this work in Applied Sciences after the following revisions.

1. In the Introduction part, more details and the latest references for PDT should be added.

2. How to regulate the permeation of ALADENT devices in bacterial membranes? Some data are needed to support this.

3. Whether will the use of SRP plus ALADENT treatment achieve better results in a short period of time? Such as: TIME 0+3 months, and even TIME 0+1 month.

4. In Table I and Table II, the author should give a clear label. Please add it.

5. In the graph I, the top right label is confused. The authors should explain that clearly in the text.

6. Several grammatical errors and typos should be corrected, and the entire manuscript should be carefully checked again.

Author Response

Ferrara, 19th February 2022

Dear editor:

Many thanks for the insightful comments and suggestions of the referees. We have

made corresponding revisions according to their advice. Words in red are the changes

we have made in the text.

  • In the Introduction part, more details and the latest references for PDT should be added. More details and the latest references for PDT were added in the Introduction part.

  • Whether will the use of SRP plus ALADENT treatment achieve better results in a short period of time? Such as TIME 0+3 months, and even TIME 0+1 months. We recorded the periodontal parameters after 6 months, in this way the obtained data showed more stable and accurate results.

  • In Table I and Table II, the author should give a clear label. Please add it. We added a clear label in Tables I and II.

  • In graph I, the top right label is confused. The authors should explain that clearly in the text. We clearly explained it in the “Microbiological Results” section: “Compared to T0 (time before treatment), total bacteria loadings have a statistically significant reduction after 15 days treatment (T0 + 15 gg) with SRP (right site) (table I + graph I) (total average decrease of 27%).”

  • Several grammatical errors and typos should be corrected, and the entire manuscript should be carefully checked again. We checked the entire manuscript, correcting grammatical errors.

Thank you for receiving our manuscript and considering it for publication.

We appreciate your time and look forward to your response.

Yours sincerely,

Prof.ssa Dorina Lauritano

Reviewer 3 Report

Ten patients (20 sites) were enrolled in a prospective clinical study to demonstrate the efficacy of ALADENT medical device. Clinical pocket depth, clinical attachment loss and bleeding on probing were evaluated at time 0 and after 6 months, while microbial analysis was conducted at baseline and after 15 days analyzed.
This topic presented are of interest, but some points need further clarification:
1.     How was the sample size determined? There is no relevant sample size calculation process in this paper. Were any samples lost during the experiment?
2.     How is the postoperative follow-up time determined? Why not follow up 1 month, 3 months after surgery?
3.     Controls for influencing variables in the study should be further elaborated. Depth of subgingival samples? The position of the tip of illuminator?
4.     Was the whole experiment conducted according to CONSORT criteria? Clinical studies need to be conducted according to CONSORT criteria, but many items were missing. Please address all sections and include a completed CONSORT checklist as supplementary material for review.
5.     Was the study protocol approved by the ethics review committee? Was the experiment officially registered? There is no content related to trial registration in this paper.
6.     There is a lack of units on some of the axes in the Figures.

Author Response

Ferrara, 19th February 2022

Dear editor:

Many thanks for the insightful comments and suggestions of the referees. We have

made corresponding revisions according to their advice. Words in red are the changes

we have made in the text.

  • How was the sample size determined? There is no relevant sample size calculation process in this paper. Were any samples lost during the experiment? We performed a sample size calculation, obtaining a series of cases. We added in the “Materials and Methods” section the description of sample size calculation.

  • How is the postoperative follow-up time determined? Why not follow up 1 month, 3 months after surgery? We recorded the periodontal parameters after 6 months, in this way the obtained data showed more stable and accurate results.

  • Controls for influencing variables in the study should be further elaborated. Depth of subgingival samples? The position of the tip of the illuminator? The subgingival samples were taken in the deepest part of the periodontal pocket and the tip illuminator was placed perpendicular to the surface of the gingival mucosa. We specified these details in the text (Clinical Methods section).

  • Was the whole experiment conducted according to CONSORT criteria? Clinical studies need to be conducted according to CONSORT criteria, but many items were missing. Please address all sections and include a completed CONSORT checklist as supplementary material for review. The research was not conducted according to CONSORT criteria, since it is not a randomized clinical trial, but it only presents a series of cases.

  • Was the study protocol approved by the ethics review committee? Was the experiment officially registered? There is no content related to trial registration in this paper. Our study doesn’t provide for an ethical committee since it was conducted in a private dental office and since the ALADENT device was already on the market when we started our research.

Thank you for receiving our manuscript and considering it for publication.

We appreciate your time and look forward to your response.

Yours sincerely,

Prof.ssa Dorina Lauritano

Round 2

Reviewer 1 Report

The authors answered to all my comments.

However, there are some bias, that affect the clinical and scientific significant of the whole study.

-"They tested 10 left sites, treating them with SRP+ ALADENT, and 10 sites, treating them with SRP alone. The differences in TBL between the sites are due to the fact that the patients physiologically brush their teeth more to one side than the other". 

-"The periodontal examination and the treatment (SRP and SRP + ALADENT) were performed by the same operator".

I am sorry, but, due to these biases, the data cannot support conclusions. I suggest the authors increase the number of patients, and invert the side TEST, in order to overstep this point.

In the current form the manuscript is to be rejected, but with the invitation to resubmit, after adding other data.

Reviewer 3 Report

The authors have fully addressed my concerns.